# Clinical Potential of Cellular Material Sources in the Generation of iPSC-Based Products for the Regeneration of Articular Cartilage

**DOI:** 10.3390/ijms241914408

**Published:** 2023-09-22

**Authors:** Artem Eremeev, Arina Pikina, Yevgeny Ruchko, Alexandra Bogomazova

**Affiliations:** 1Lopukhin Federal Research and Clinical Center of Physical-Chemical Medicine, Federal Medical Biological Agency, Malaya Pirogovskaya 1a, Moscow 119435, Russia; info@rector.msu.ru (A.P.); abogomazova@rcpcm.org (A.B.); 2Koltzov Institute of Developmental Biology, Russian Academy of Sciences, 26 Vavilov Street, Moscow 119334, Russia; info@idbras.ru; 3Department of Embryology, Faculty of Biology, Lomonosov Moscow State University, GSP-1 Leninskie Gory, Moscow 119991, Russia

**Keywords:** induced pluripotent stem cells (iPSCs), joint diseases, cell technologies, chondrogenesis in vitro

## Abstract

Inflammatory joint diseases, among which osteoarthritis and rheumatoid arthritis are the most common, are characterized by progressive degeneration of the cartilage tissue, resulting in the threat of limited or lost joint functionality in the absence of treatment. Currently, treating these diseases is difficult, and a number of existing treatment and prevention measures are not entirely effective and are complicated by the patients’ conditions, the multifactorial nature of the pathology, and an incomplete understanding of the etiology. Cellular technologies based on induced pluripotent stem cells (iPSCs) can provide a vast cellular resource for the production of artificial cartilage tissue for replacement therapy and allow the possibility of a personalized approach. However, the question remains whether a number of etiological abnormalities associated with joint disease are transmitted from the source cell to iPSCs and their chondrocyte derivatives. Some data state that there is no difference between the iPSCs and their derivatives from healthy and sick donors; however, there are other data indicating a dissimilarity. Therefore, this topic requires a thorough study of the differentiation potential of iPSCs and the factors influencing it, the risk factors associated with joint diseases, and a comparative analysis of the characteristics of cells obtained from patients. Together with cultivation optimization methods, these measures can increase the efficiency of obtaining cell technology products and make their wide practical application possible.

## 1. Introduction

Inflammatory diseases of the joints are characterized by degenerative changes in cartilage tissue and have a chronic course. The most common diseases of cartilage tissue are osteoarthritis (OA) and rheumatoid arthritis (RA), which have different etiologies but ultimately both lead to cartilage destruction [1]. Avascularization, a low proliferative activity for mature chondrocytes, and a large amount of extracellular matrix (ECM) in combination with neglected pathogenetic mechanisms prevent the natural regeneration of articular cartilage; as a result, the diseases progress to completely degenerate the articular apparatus, involving the surface layers of bone and cartilage tissue [1,2]. According to modern concepts, arthrosis results from the interaction of age-related, hormonal, inflammatory, immunological, genetic, and environmental factors leading to cellular stress and the degradation of the ECM that occur with macro- and microdamage [3]. The main risk groups for the occurrence of OA are mainly the elderly, prone to fullness or performing activities associated with injuries and heavy stress on the joints [4]. Thus, the pathogenesis is based on the predominance of catabolic processes over anabolic processes, particularly due to the inadequate reparative response, including the activation of pro-inflammatory immune system pathways [5].

Pathological processes at the molecular level occur simultaneously in cartilage tissue, subchondral bone, menisci, and the synovial membrane, leading to structural changes and the loss of biological properties for all joint tissues: cartilage degradation, bone remodeling, formation of osteophytes and chondrophytes, inflammation, and edema. Subsequently, capsule-ligamentous structures and muscles surrounding the joint are involved in the pathological process [6].

In accordance with clinical recommendations, both non-surgical approaches (such as conservative treatment) and surgical methods are used in the treatment of gonarthrosis or degenerative-dystrophic knee joint disease patients, depending on the stage of the disease [7]. Non-surgical approaches include pharmacological and non-pharmacological approaches. As for non-pharmacological approaches, they include patient training, body weight correction, exercise therapy, orthopedic devices, and physiotherapy methods [8,9]. Pharmacological methods include the symptomatic use of nonsteroidal anti-inflammatory drugs (NSAIDs), short-term intra-articular administration of glucocorticosteroids or high-molecular derivatives of hyaluronic acid. Surgical methods of treating gonarthrosis include arthroscopy of the knee joint, tunneling and microfracturing, abrasive chondroplasty, corrective periarticular osteotomies of the femur and/or tibia, and knee replacement [10,11,12].

Non-surgical treatment is used for stages I–II, whereas surgical treatment is prescribed for stages II–III or in cases where conservative therapy is ineffective in patients at stage I of the pathological process [13]. Pharmacological methods using NSAIDs, from paracetamol to potent drugs, are characterized by highly likely side effects [14]. Additionally, the use of glucosamine and chondroitin is not recommended for gonarthrosis patients [15]. Although intra-articularly administrating high-molecular hyaluronic acid derivatives can lead to a positive clinical effect, its severity and duration vary strongly depending on the physicochemical characteristics of a particular drug [16]. Intra-articular injection of glucocorticosteroids has a minimal and short-term therapeutic effect, therefore, given the high risk of secondary osteonecrosis and the destructive impact on articular cartilage, these substances are not recommended for gonarthrosis patients [17,18]. In other words, all types of treatment are currently symptomatic, as drugs that restore the morphology and functional activity of hyaline cartilage are completely absent [19,20]. This increases the relevance of developing technologies aimed at correcting the causes of joint function loss. So far, the most radical and effective way to treat this pathology is knee replacement [21,22,23]. This surgical intervention is an operation in the high complexity category and is recommended for patients aged 50 years and older [24,25].

The use of biomedical cell products is considered more justified in patients at stages I-II of the pathological process [26,27,28,29,30]. With severe deformities at stage II of the disease, as well as at stage III, it is necessary to perform a corrective osteotomy and, in more severe cases, a knee joint prosthesis [31,32].

It is also necessary to separately note a significant social group with a high risk of arthrosis—athletes, military personnel, and other persons with a high risk of injury due to their kind of activity [33]. Considering the disease’s traumatic etiology, their relatively young age, as well as their small number of comorbidities, patients who are actively involved in sports could be an important target group for the use of cellular products, as delaying arthroplasty is important for their professional career [34].

To date, the most commonly used resource for cellular technologies for cartilage regeneration are mesenchymal stem cells (MSCs) [35,36]. Directly or with the application of biomaterials, MSCs have shown efficacy in animal models [37,38,39,40] and in treatment [41,42,43]. However, the use of MSCs for cell therapy has a number of disadvantages, such as heterogeneity, limited expansion in vitro, and a high risk of hypertrophy of the resulting cartilage tissue [44,45,46]. Autologous chondrocytes are another important and proven cellular resource in clinical practice [47,48]. Implantation of autologous chondrocytes can also be carried with the help of various bioscaffolds; in a number of clinical cases this has demonstrated medium-term efficacy [30,49,50]. Despite the fact that cellular technology products based on autologous chondrocytes for cartilage repair are already present on the medical and biotechnological markets, so far, they are aimed primarily at correcting minor damage to the articular cartilage and do not cover all the needs of patients with serious joint pathologies or malformations [22,51,52]. In particular, with the appearance of osteophytes and, in severe cases, joint and limb deformities, the use of cellular technologies may be limited. Moreover, often due to a number of reasons—for example, in the case of prolonged glucocorticoid treatment, radiation, and chemotherapy, when the chondrogenic potential of cartilage tissue is lost—it is impossible to obtain enough autologous chondrocytes to create a full-fledged cartilage implant, which requires a large number of chondrogenic differentiated cells [53,54].

In this regard, induced pluripotent stem cells (iPSCs) are a promising source for obtaining various differentiated derivatives, including chondrogenically differentiated cells [55]. Unlike autologous chondrocytes and MSCs, iPSCs demonstrate unlimited growth in vitro and a wide range of origin tissues, meaning they have similar properties to embryonic stem cells (ESCs) but are devoid of ethical problems [56]. These advantages for iPSCs make it possible to obtain an extensive cellular source for further manipulations. However, data on the safety of iPSCs and the ability to form a full-fledged functionally active cartilage tissue are quite limited, as well as contradictory with regard to immunogenicity. Recent advances in the field of articular cartilage regenerative medicine using iPSC-based constructs make it possible to consider using such tissue-engineered constructs as a promising approach to replacement therapy and restoring joint cartilage tissue function [57,58,59]. The personification of regenerative medicine favors the autologous approach when creating biomedical cellular products and technologies based on differentiated iPSC derivatives for a number of reasons. Firstly, autologous iPSC lines make it possible to obtain patient-specific chondrogenic constructs that do not cause immune rejection without the need for additional immunosuppressive manipulations. Secondly, with such an approach and due to the observance of good manufacturing practices, the need for a routine process for infectious agent screening is eliminated.

To date, several transplantations of autologous cells differentiated from iPSCs have been reported without serious consequences threatening the health of patients and without the need for immunosuppressive therapy [60,61,62,63]. In addition to regenerative medicine, differentiated chondrogenic derivatives obtained from the iPSCs of patients with cartilage tissue pathologies can be used to study the complex mechanisms of multifactorial diseases of articular cartilage in vitro, as well as to develop new diagnostic methods.

At the same time, obtaining autologous iPSCs is a long and expensive process that makes it difficult to introduce cell technology products based on this stem-cell type into medical practice. Imperfect differentiation protocols do not yet allow one to obtain mass cultures of chondrocytes without the risk of heterogeneity. There are also publications on the reduced chondrogenic potential of differentiated iPSC derivatives obtained from osteoarthritis patients compared with iPSCs from healthy donors, which carries the risks of obtaining a cellular product with potentially less therapeutic efficacy [64,65,66,67].

In the case of hereditary pathologies, the resulting iPSCs can inherit etiological factors that can affect the properties of the resulting differentiated cells. In a study in which iPSCs were obtained from the fibroblasts of a patient with a lethal variation of skeletal dysplasia, a reduced and aberrant secretion of the cartilage ECM was observed and, in contrast to iPSC lines obtained from the somatic cells of a healthy donor, using BMP2 and TGFβ did not have a positive effect on the degree of differentiation [68]. Human dermal fibroblast iPSCs from type I thanatophoric dysplasia and achondroplasia patients differentiated into chondrocytes with low expressions of glycosaminoglycans and the FGF3 receptor [69]. Additionally, chondrogenic iPSC derivatives obtained from the MSCs of bone marrow [70] and the dermal fibroblasts [71] of osteochondritis dissecans patients were associated with defective aggrecan processing and, as a result, reduced production of ECM. At the same time, this pathological cartilage phenotype was also manifested in the teratoma chondrogenic area after transplanting the mouse iPSC line. These data demonstrate that iPSCs and their derivatives obtained from patients with hereditary chondrogenic anomalies display impaired chondrogenesis at the cellular and molecular levels, making them impossible to use in clinical practice. However, the diseases listed above are caused by monogenic mutations. It remains to be seen what the situation will be with the cells of patients whose cartilage diseases are caused by environmental factors, in combination with a multifactorial genetic predisposition.

In this review, we present data on iPSCs obtained from the cells of donors with various multifactorial diseases of the articular cartilage. We will discuss the phenomenon of epigenetic memory, its role in in vitro differentiation and pathological phenotype inheritance, as well as experimental data on the characteristics of iPSCs and their chondrogenic derivatives from joint disease patients and some known molecular factors of joint pathologies. In addition, we offer a compilation of some recommendations for optimizing work with iPSCs from donors with joint diseases in vitro.

## 2. Differentiation Potential and Epigenetics of IPSC

There is some evidence that somatic cells of various origins demonstrate differential susceptibility to the reprogramming procedure. This is due to juvenescence, namely the difference in the degree of maturity, and the potency of a particular cell type to differentiate. For example, neonatal keratinocytes reprogram faster and more efficiently than adult cells [72]. Moreover, compared with chondrocytes, fibroblasts undergo the process of reprogramming more easily [73]. In addition, iPSCs of different origin differentiate into the target cell type with unequal efficiency. Initially, as an explanation for these phenomena, researchers promoted the concept of epigenetic memory, or the ability of iPSCs to inherit the molecular features inherent in their original cells. As such, iPSCs should contain residual epigenetic marks that prevent commitment to a certain cell fate or, conversely, have no marks that promote the development of this cell fate [74]. Experiments have shown that iPSCs derived from dermal fibroblasts showed a greater propensity to differentiate into connective tissue cells compared with iPSCs from blood cells. Comprehensive high-throughput arrays for relative methylation (CHARM) revealed differentially methylated regions unique to fibroblast iPSCs and blood cell iPSCs, with the more expressed regions accounting for osteogenesis or hematopoiesis genes, respectively [74]. Later, the differentiation potential and methylation profiles of iPSCs obtained from umbilical cord blood cells and neonatal keratinocytes were compared. iPSCs from keratinocytes differentiated more easily and efficiently into keratinocytes and iPSCs from blood cells into myeloid cells. With the help of CHARM, it was also found that despite meeting the generally accepted criteria of pluripotency, both iPSC lines had significantly different methylation patterns [72]. In 2014, Boreström and co-authors conducted a comparative study in which iPSCs were obtained from the chondrocytes and fibroblasts of OA patients by a non-integrative method of delivering modified synthetic mRNAs into cells in order to minimize the genomic aberrations during reprogramming. Then, the obtained iPSCs were differentiated into chondrocytes under 3D cultivation conditions. As a result, despite the fact that reprogramming was faster and more effective for fibroblasts, chondrogenic differentiation was better for chondrocyte-obtained iPSCs, as evidenced by a higher level of expression of the ECM [73]. Although assessing methylation patterns is widely conducted to study the characteristics of iPSC lines and their derivatives, this is not the only factor that contributes to the maintenance of epigenetic memory. Thus, it was shown that differences in the methylation patterns in iPSC lines of different origin were less than the final differences in gene expression [75]. Together, these data show that residual epigenetic memory is a significant factor for the technology of obtaining standardized iPSC lines, as it affects the subsequent process of differentiation in vitro. Due to the fact that original cell type epigenomes preserved in iPSCs can direct the choice of cell fate towards the original somatic cells, the expediency of using biopsy material cells to obtain iPSCs corresponding to the direction of further differentiation was discussed [76].

However, the problem of residual phenotypes and differences in the characteristics of iPSCs and their derivatives was elaborated in further detail. A series of papers by other authors studying DNA methylation and analyzing the expression of IPSC genes obtained from donor blood and fibroblasts showed that there is a tendency for donor-specific clustering patterns and emphasized the minimal contribution of cellular origin to genetic and epigenetic differences [77]. In a study of the ability of various somatic cell iPSCs to differentiate into hepatocytes, it was found that the differentiation efficiency was lower in iPSC derivatives from Parkinson’s disease donor cells compared with healthy donor cell derivatives, regardless of the type of initial cells [78]. Another group of researchers also highlighted a significant shortcoming of previous epigenetic memory studies, which is that the contributions of donor genetic identity and the residual phenotypes are not fully separated. In this study, the main components of principal component analysis (PCA) were analyzed in addition to hierarchical clustering, which similarly did not reveal how fibroblast and lymphoblastoid cell iPSC variability depends on cellular origin. At the same time, several differential methylation areas between the experimental groups of cells associated with the origin were recorded, but the effect of these differences remains low [79]. Other research reveals the high stochastic contribution to the transcriptional and epigenetic variations of iPSCs, preventing the preservation of the original somatic cells’ specific features [80]. In sum, these studies provide a general conclusion that genetic differences between donors and a stochastic factor, rather than cellular origin, make a greater contribution to the differences in the effectiveness of reprogramming and differentiation. Nevertheless, studies on the phenomenon of epigenetic memory still appear and stimulate discussion [81,82]. Thus, in the search for a potential cellular resource for cell therapy in Huntington’s disease, researchers considered ganglionic eminence cells for reprogramming into iPSCs and further differentiation into striatal progenitors, since in the study of methylome, analysis of gene enrichment showed differences in the areas of demethylation with pluripotent cells associated with striatal genes [82].

According to the data on epigenetic memory, it can also be assumed that pathological cell iPSCs can inherit epigenetic patterns characteristic of a particular disease. Annual OA reviews shed light on more and more epigenetic markers that are identified as those of multifactorial polygenic OA, plenty of which may also serve as therapeutic targets. Among them are characteristic miRNAs, circular RNAs, and long non-coding DNAs, as well as patterns of methylation and hydroxymethylation. Since most of the genetic variations associated with OA are in non-coding regions of the genome, gene regulation plays a large role in the development of pathology, which means that epigenetics will be the focus for researchers [83,84]. However, there are practically no studies devoted to the search for these epigenetic markers in iPSC cultures obtained from articular cartilage disease patients. In work on the development of a new line of iPSCs from OA donor MSCs, Pichard et al., note that iPSCs can become an important tool in the study of the genetic and epigenetic factors of OA [81]. Finally, in a recent study, Khan et al., demonstrated the dissimilar differentiation potential of iPSCs derived from healthy and OA chondrocytes, determined by epigenetic and metabolic factors. In this study, iPSCs from healthy chondrocytes were more successful in directed chondrogenesis in vitro than iPSCs from OA chondrocytes. This was manifested in better expression and production of the chondrogenic markers SOX9, COL2A1, ACAN, and PRG4. Gene Ontology and KEGG analyses showed that in differentiated chondrocytes from iPSCs from healthy donors, the most enriched processes are epigenetic regulation, histone modification, and chromatin organization, which could contribute to a better chondrogenic potential. In addition, even in the undifferentiated state, the expression levels of such epigenetic regulators as FOXM1, IRF3, FOXP1, MYBL2, MYBBP1A, HDAC10, HDAC11, ARID4B, BRD4, HDAC4, HDAC9, KDM5A, and others were higher in iPSCs from healthy cells. These data allow us to conclude that the regulation of epigenetic modifiers may have a positive effect on chondrogenesis ability and residual epigenetic memory may be one of the most important causes of the pathological state of cells [85]. In addition, there is evidence that a prolonged inflammatory process provokes somatic mutagenesis in tissues. So, in ulcerative colitis, mutations accumulate in many intestinal epithelium genes associated with signaling [86]. Therefore, the risk associated with the preservation of pathological features in the genetic and epigenetic landscapes must be considered and studied (Figure 1).

## 3. iPSCs Derived from the Tissues of Patients with Articular Cartilage Pathology

iPSCs for chondrogenic differentiation are derived from a variety of cell types. Using different differentiation protocols, the researchers obtained chondrocyte-like cells from iPSCs of diverse origin, and the resulting cultures demonstrated the production of the cartilage matrix and other characteristic markers [64,65,66,87,88,89,90]. However, there are few studies comparing iPSCs obtained from the tissues of articular cartilage disease patients and healthy donors to date [64,65,67,73,81,88].

Most studies provide data on the successful generation of iPSC lines from various cell types of joint disease patients according to standard criteria for pluripotency [64,65,66,87,88,89,90]. It was reported that, according to the characteristics of pluripotency marker expression and the ability for unlimited expansion in vitro, iPSCs obtained from dermal fibroblasts [64], chondrocytes [65], MSCs [87], synovial cells [66,87,88], and blood cells [89,90] from donors with articular cartilage diseases do not differ from healthy lines. Thus, it was possible to obtain iPSC lines from the fibroblast-like synovial cells of patients using a lentiviral vector, and the researchers stated that a line of patient-specific iPSCs from donors with RA was obtained for the first time. iPSCs f RA and OA patients demonstrated the expression of pluripotency markers and effectively differentiated into osteoblasts [87]. In another study, both fibroblast-like synovial cells and peripheral blood mononuclear cells (PBMCs) from donors with RA were successfully reprogrammed into iPSCs [90]. iPSCs were also obtained from the PBMCs of ankylosing spondylitis patients using episomal vectors [89]. The results of these studies allow us to consider the cells of joint disease patients as a cellular resource for reprogramming, and the resulting iPSCs can be used for chondrogenic differentiation. However, it should be noted that the authors do not provide a detailed analysis of the obtained cells’ epigenomes and transcriptomes or the assessment of genome modifications as a result of the reprogramming procedure, nor do they evaluate the chondrogenic potential of such iPSCs. Meanwhile, some researchers have been able to effectively differentiate the iPSCs derived from cells of pathological tissues. In a study by Kim et al., in 2011, iPSC lines derived from the synovial cells of OA patients demonstrated the characteristic expression profiles of pluripotency genes and were successfully differentiated along the chondrogenic pathway by directed differentiation in vitro. The resulting chondrogenic derivatives produced proteoglycan and collagen ECM and expressed SOX9, whereas a marker such as type X collagen was found only in one of the three lines obtained from cartilage disease patients [88]. The study by Zhu et al., also reported no defects in either the iPSC line derived from OA patient dermal fibroblasts or chondrogenic derivatives. In addition, transplantation of a suspension of chondrocyte-like cells into osteochondral defects in rats caused an increase in the production of proteoglycans and type 2 collagen and proliferation of chondrocytes; however, regeneration was incomplete and the cartilage was not fully restored [57]. Thus, obtaining iPSCs from patients with cartilage pathologies and special conditions for cultivation to obtain differentiated iPSC derivatives overcomes the problem of low proliferative activity and reduced potential for differentiation in the production of autologous products based on chondrocytes.

However, there is also conflicting evidence that iPSCs derived from pathological cells exhibit delayed proliferative activity and a lower quality of differentiation; such altered biological properties are usually attributed to the original cells that underwent reprogramming [87]. One study found that chondrocytes differentiated from such iPSCs showed less ability to produce cartilage ECM compared with healthy donor cells [64]. Other authors note that differentiated chondrocytes from iPSCs from donors with OA express a number of chondrogenic markers, such as COL2, ACAN, and COMP, at a high level. However, at the same time, VEGF expression was recorded, which indicates an abnormal direction of differentiation [65]. Additionally, chondrocytes obtained from iPSCs that were also obtained from osteoarthritis cartilage cells not only demonstrated a tendency for chondrogenic and osteogenic differentiation in 3D pellet cultures but also had high expression of the COL10A1 hypertrophic marker, as did donor cells, which researchers associated with the preservation of the pathological phenotype’s epigenetic memory [73].

Some studies that consider iPSCs from donor cells with various joint diseases as material for in vitro pathology modeling also emphasize the inheritance of abnormal cellular characteristics by chondrogenic derivatives. Rim et al., obtained iPSCs from the fibroblast-like synovial cells (FSCs) of RA patients using a lentiviral system and then differentiated them into chondrocytes. The resulting clones were more prone to mineralization during osteogenic differentiation than clones derived from healthy control cells. However, it should be considered that FSCs are most exposed to the inflammatory process inside the synovial joint and acquire a tumor-like phenotype, thereby contributing to disease progression. Therefore, compared with other cells, FSCs probably best reflect the state of the patient’s pathologically affected cartilage tissue [66]. The same group of authors, in study [67], made another important observation for the chondrogenic derivatives obtained from IPSC fibroblasts from the skin of an early finger OA patient. The expression levels of OA markers, such as COL1A1, RUNX2, AQP1, VEGF, IL-6, MMP1, and 10, were significantly higher in iPSC derivatives from OA cells compared with the control, but significant differences disappeared in final differentiated cells. However, type II collagen secretion was still reduced. Thus, currently we have slightly scattered results on the properties of iPSCs and their derivatives obtained from donor cells with joint pathologies (Table 1).

## 4. Risk Factors in iPSCs Obtained from Articular Cartilage Disease Patients

The etiology of different arthritis forms is multifactorial and both genetic and external factors contribute to the risk of developing the disease. At the same time, risk factors may be present not only in patients with severe symptoms of the disease but also in healthy people with a predisposition, albeit at a lower frequency [92]. Arthritis is a whole family of diseases with fundamentally different mechanisms underlying the development of pathology, and it is quite difficult to identify these mechanisms. Despite this, it is very important to investigate the pathogenesis of articular cartilage diseases for modeling and screening purposes. However, in order to use iPSCs from donors with cartilage pathology, it is essential to understand whether a specific genetic or metabolic factor is associated with the final phenotype of differentiated chondrocyte derivatives and what effect this may have on therapeutic practice.

Genome-wide association studies (GWAS) have shown that multiple single nucleotide polymorphisms (SNPs) can be associated with OA pathogenesis. Thus, SNP in the GDF5 gene—which is responsible for stimulating anabolic and inhibiting catabolic enzymes during chondrogenesis—was found in patients with hip and knee joint OA [93]. In iPSCs obtained from the cells of OA patients, SNP brushes contributing to the development of pathology were found in the GDF 5, SMAD3, ALDH1A2, and IL1-R1 genes involved in the transmission of signals by growth factors [64]. It was also found that SNPs in the genes of collagen XI, VEGF, GDF5, and IL8 are associated with OA, although this relationship is not always consistent due to the multifactorial nature of the disease [94]. In addition, SNPs are an important diagnostic tool since polymorphisms can be specific to subtypes of arthritis. For example, mutations in COL 11 A and VEGF were associated with hip joint OA, whereas those in COL9A3, ASP, DVWA, and GDF 5, were associated with knee joint OA [94].

Genome-wide sequencing the exomes of families suffering from hereditary cases of OA with extended early onset and without dysplasia identified a mutation in the collagen-binding domain of fibronectin that leads to a decrease in type II collagen binding. To study the functional role of this mutation in chondrogenesis, a model using the CRISPR-Cas9 method and IPSCs was also developed. The obtained 3D cartilage models showed a decrease in the expression of type I and II collagens, fibronectin, and aggrecan, alongside an increased marker expression of progressive OA ADAMTS-5, ALPL, and RUNX2 [95]. These data allow us to consider this mutation as one of the main causes of primary human OA observed at ages of 20–40 years, but its contribution to other cases of OA is currently unknown. Additionally, in MSC-like derivatives of iPSCs obtained from the dermis fibroblasts of axial spondyloarthritis patients, increased expressions of the genes predisposing this disease—EDIL3, ANO6, HAPLN1, and ANTXR2—were detected, which also provided a prerequisite for effective diagnosis and study of the pathogenesis using functional genomics [96]. Ankylosing spondylitis is also a complex multifactorial joint disease; however, its main risk factor (which is present in most patients) is the HLA-B27 variant. It was possible to obtain an iPSC line from HLA-B27 carrier donor cells that corresponded to the basic necessary characteristics of pluripotent cells, although this line’s ability to chondrogenically differentiate was not evaluated [89].

The study of metabolomics can also be useful for a better understanding of pathogenesis. A comparison of the IPSC metabolomic profiles obtained from OA and RA patient chondrocytes, as well as fibroblast-like synoviocytes, showed a significant difference in the metabolites of donor cells with different diseases. Greater amounts of adenine and NAM were observed in RA cells, which firstly opens up diagnostic possibilities at the molecular level and secondly suggests considering NAM inhibitors—for example, tannic acid—as potential therapeutic agents for the treatment of RA [97]. Studies with IL-6 and MP-1 and 10 knockdowns in iPSCs obtained from cells of OA patients would help assess the contribution of these factors to cartilage degradation [67].

Another factor contributing to the initiation and progression of OA is the cellular senescence of chondrocytes, which is especially aggravated with aging [98,99,100]. The phenomenon of cellular aging includes many processes associated with slowing down or stopping the cell cycle and changes in morphology and secretory activity [101]. Among the main features, telomere depletion and genetic mutations, including those in the mitochondrial genome, can be noted as various epigenetic marks specific to aging [102]. All these factors together change cellular activity, which has an effect on the development of joint disease and its course. Moreover, senescent chondrocytes produce metalloproteinases and cytokines that are involved in the development of OA and are markers of this disease [20,103]. Considering that OA often affects the elderly, it is necessary to study the risk of a negative effect of the senescent phenotype of the original cells on the properties of iPSCs and their derivatives. There are conflicting data on the efficiency of reprogramming cells from elderly donors. Some studies report a reduced efficiency for the reprogramming process compared with cells from younger donors [104,105], but others have demonstrate successfully obtaining iPSCs even from long-lived donor cells [106,107]. What these studies have in common is the lack of coverage of donor specificity and the comparison of small numbers of iPSC lines, so further comparisons that account for these factors are needed going forward. However, the current consensus is that the signs of aging in cells are erased as a result of reprogramming [108]. As a result, iPSCs acquire a juvenile phenotype comparable with ESCs, with elongated telomeres and without the epigenetic markers of aging [106,108,109]. Moreover, iPSC exosomes can have a leveling effect on mature somatic cells [110]. However, despite this, there are studies where iPSCs are considered as an in vitro model for studying cellular aging under conditions of long-term cultivation [111,112]. Therefore, signs of aging may appear again in iPSCs after longer cultivation. In any case, this problem requires further study.

## 5. Approaches to Work with iPSCs Obtained from Cells of Cartilage Disease Patients, as Well as Their Differentiated Derivatives

Introducing cellular technologies using autologous UCS into the medical practice surrounding joint pathology therapy is possible only after safety studies in preclinical and clinical trials. As a rule, iPSCs themselves are not used to create cellular or tissue-engineered drugs due to the high risk of teratoma formation, but their differentiated derivatives are promising (Table 2). Even at the stage of obtaining clones, iPSCs must undergo strict control, which allows us to weed out the resulting clones with various morphological, karyotype, etc., disorders [61]. The mandatory studies of iPSC derivatives before clinical approbation test for oncogenicity [113,114], teratogenicity [115,116], genomic aberrations [117], and the heterogeneity of the final differentiated cultures [117,118]. In addition, to make future therapeutic use of differentiated iPSC derivatives safer, it is necessary to optimize the production and cultivation technologies, since they also affect the quality of a potential cellular drug.

It is known that lentiviral or retroviral integration during reprogramming can make changes to the genome such as insertion damage or chromosomal aberrations [92], as well as remove the epigenetic marks inherent in the original cells [73]. In the future, these factors may affect the quality of further iPSC differentiation and the characteristics of final chondrogenic derivatives, which may not allow the use of such cells in clinical practice. Currently, many researchers prefer non-integration methods, such as reprogramming using the Sendai virus [64], episomal vectors [119], and mRNA [73]. Ultimately, improved reprogramming techniques can generate iPSCs that better match the genetic and epigenetic features of pluripotent stem cells. In addition to excluding the viral integration factor, non-integrative methods can also contribute to more effective preservation of epigenetic and transcription profiles, which may be essential for obtaining iPSCs from donor chondrocytes and their reverse differentiation into chondrocytes when creating a cellular preparation [73]. It is noteworthy that some cell properties of patients with joint pathologies that differ from normal ones can also act as advantages in reprogramming. For example, the synovial cells of RA patients express Klf4 at sufficient levels to avoid using this factor during reprogramming. This makes it possible to minimize the use of this pluripotency factor, thereby reducing the risks of oncogenicity [87].

**Table 2 ijms-24-14408-t002:** Recent advances in chondrogenic differentiation of iPSCs and models in vivo.

Reprogramming Method	Type of Reprogrammed Cells	Method of Chondrogenic Differentiation	Matrix	Characteristics of Chondrogenic Derivatives	Model In Vivo, Procedure	Transplantation Results	Link
Minicircle vector	Human fibroblasts, adipose-derived stem cells	Differentiation through the stage of MSC-like precursors using dexamethasone, ascorbic acid and TGFβ3 in 3D high-density pellets culture.	Polyethylene glycol (PEG) and chondroitin sulfate methacrylate (CSM) based scaffold.	On the 14th day of differentiation, cartilage markers COL2A1, COL9A1, COL11A1, SOX9, and ACAN were expressed in derivatives. The expression of a marker of hypertrophy COL10A1 and a marker of fibrosis COL1A2 was also recorded. On day 21 of differentiation, alcian blue staining revealed the presence of proteoglycans and there was also positive immunostaining for type II collagen.	Athymic nude Sprague Dawley rats, transplantation of 21-day 3D-pellets into osteochondral defects of knee joints	At 6 weeks after transplantation, a significant decrease in the relaxation time T2 of grafts was observed, which indicates their dehydration and matrix production. Hematoxylin and eosin staining showed engraftment of cell grafts. Positive staining with alcian blue and immunochemical staining for type II collagen demonstrated remodeling of the defect, whereas the control group of empty scaffolds had no effect. No tumors or teratomas were found.	[120]
Episomal vectors	Human dermal fibroblast	Chondrocyte-specific iPSC reporter lines were created by introducing the COL11A2-EGFP human transgene. Differentiation through the stage of MSC-like precursors using Wnt3a and Activin A. Then, differentiation was carried out using ascorbic acid, BMP2, TGFβ1, and GDF5; after 14 days of cultivation, there was a transfer to a suspension culture.	-	In adherent culture, cells formed nodules that specifically showed COL11A2-EGFP fluorescence on day 14 of differentiation and almost all cells expressed COL11A2-EGFP on day 56 in suspension culture. The expression levels of chondrogenic markers SOX9, COL2A1, COL11A2 increased with differentiation. The proportion of SOX9-positive cells by the 42nd day of cultivation reached 91.8% ± 0.91%. On the 28th day of differentiation, slight staining with safranin O was observed but by day 42 it became intense. Immunohistochemistry showed the presence of both type I and type II collagen. Expression of collagen type I was reduced by manipulating the composition of the medium. IHH and COL10A1 mRNA expression levels were lower than in native cartilage, indicating low hypertrophy.	SCID mice, subcutaneous transplantation of 42-day cell constructs SCID rats, transplantation of 28-day cell constructs into osteochondral defects of knee joints Mini-pigs, transplantation of 56-day cell constructs into osteochondral defects of knee joints	Hyaline-like cartilage formation after subcutaneous transplantation with high collagen type II expression and intense safranin O staining and low expression of collagen types I and X. Twelve months after transplantation, collagen X expression and epiphyseal-like cartilage were observed in some areas, suggesting hypertrophy. After transplantation into defects of the knee joint of both rats and minipigs, extensive integration into the cartilage, positive staining for safranin O. In the case of rats, also intense staining with toluidine blue and the presence of type II collagen were observed. Cell clusters did not cause the formation of tumors and ectopic tissue damage as a result of transplantation.	[121]
(not specified)	Dermal fibroblasts of patient with knee OA	Directed differentiation in EBs using ascorbic acid, dexamethasone, TGFβ1 for 2 days, then the EBs were transferred onto cultural plastic coated with gelatin and differentiation continued in the same medium.	-	After 14 days of differentiation, intense toluidine blue staining and expression of chondrogenic markers COL2A1, ACAN, and SOX9 were observed.	Sprague Dawley rats, transplantation of cell suspension into osteochondral defects of knee joints	Fifteen weeks after transplantation, an increase in the content of proteoglycans, type 2 collagen, as well as proliferation of chondrocytes was recorded. However, the amount of cartilage matrix in the damaged area did not reach that in the healthy joint. The improvement in joint function reduced lameness in rats, but the cartilage was not completely restored. No tumors or teratomas were found	[57]
Sendai virus	Human cord blood mononuclear cells (CBMCs)	Directed differentiation in EBs seeded on gelatin-coated plastic, using ascorbic acid, dexamethasone, BMP2, and TGFβ3. After 4 days of cultivation the culture was transferred to 3D pellet conditions.		After 30 days of differentiation high levels of expression of chondrogenic markers SOX9, ACAN, and COL2A1 in pellets were observed; however, the levels of hypertrophic marker COL10A1 and fibrosis marker COL1A1 were also high. At the same time, the level of expression of type I collagen was higher than that of type II collagen, whereas the data for protein production were opposite. The pellets were also positively stained with toluidine blue.	Sprague-Dawley rats, transplantation of 30-day 3D pellets into osteochondral defects trochlear groove of the distal femur	At 8 weeks after transplantation, intense staining with toluidine blue and safranin O was observed in the area of the defect, demonstrating proteoglycan production and normally organized cartilage morphology. Cells inside the pellet formed lacunae. Compared with the introduction of a suspension of the obtained differentiated derivatives, the pellets showed a better therapeutic effect, although the suspension also contributed to the restoration of the cartilage. No tumors or teratomas were found.	[58]
Episomal plasmid vectors without transgenes	Mouse embryonic fibroblasts	Differentiation through the stage of MSC-like precursors using fetal bovine serum (FBS) and bFGF. Then, differentiation was carried out in high-density micromass culture or alginate gel using BMP2.	Ultra-purified alginate gel	After 28 days of differentiation, alcian blue staining was intense, both in the culture within alginate gel and in micromass culture. Expression levels of the chondrogenic markers SOX9, COL2A1, and ACAN were high in both gel culture and micromass culture and increased during differentiation. Expression of the osteogenic markers Runx2, ALP, COL10A1 and adipogenic marker PPARγ increased only in high-density micromass culture.	Nude mice BALB/cScl- nu/nu, transplantation of cell suspension into gel into dorsal flanks	On the 28th day after transplantation, intense alcian blue staining was observed, as well as immunostaining for type II collagen. Additionally, over time, the expression of COL2A1 and ACAN mRNAs increased, whereas the expression of SOX9 remained almost constant. No tumors or teratomas were found.	[122]
Sendai virus	Normal human epidermal keratinocytes (NHEK)	Differentiation through the stage of MSC-like precursors. Then, differentiation was carried out using TGFβ1 and ascorbic acid.	-	After 26 days of differentiation, staining of micromasses with hematoxylin and eosin showed cartilaginous morphology, intense staining with safranin O and immunostaining for aggrecan and type II collagen were also recorded.	New Zealand white rabbits, transplantation of cell suspension into knee osteochondral defect	Twelve weeks after transplantation, intense safranin O staining and aggrecan immunostaining were observed. Histological evaluation of ICRS scores demonstrated a significant superiority for cartilage histology after transplantation compared with untreated controls. A decrease in the expression of markers of inflammation and catabolism IL-1β, TNF- α, and MMP13 was also observed. No tumors or teratomas were found.	[123]
(not specified)	Mouse gingival fibroblasts	Directed differentiation via 3D pellet formation with BMP-4, then with BMP4, dexamethasone, and TGFβ3 on a 3D orbital shaker.	-	After 28 days of differentiation in a rotational suspension culture the pellets acquired the appearance of a hyaline-like cartilage and were positively stained with safranin O. High levels of expression of the chondrogenic markers SOX9, ACAN, and COL2A1 were also recorded. Immunostaining for type I collagen was slight.	Sprague-Dawley rats, transplantation of 28-day pellets into the superficial osteochondral defects	Four weeks after transplantation, the filling of the defect with tissue similar to cartilage was observed and microCT images showed complete repair of the tissue and full integration of pellets. The healing area was stained intensely with safranin O and showed high production of type II collagen and low levels of type I and X. Signs of the tumor formation of pellets were not detected.	[124]
(not specified)	Human CBMCs	Directed differentiation through the stage of EBs that were resuspended and seeded on gelatin-covered plastic. Dexamethasone, BMP2, and TGFβ3 were used.	-	After 14 days of differentiation, staining with safranin O, toluidine blue, and alcian blue showed accumulation of cartilage matrix. High levels of expression of chondrogenic markers SOX9, ACAN, and COL2 were also recorded, comparable with those in primary chondrocytes. Large amounts of type I and II collagens and fibronectin were recorded in the decellularized ECM.	Sprague-Dawley rats, transplantation of decellularized ECM into osteochondral defect on the articular cartilage of the trochlear sulcus of the distal femur	Twelve weeks after transplantation, high accumulation of cartilage matrix, in particular, collagen type II after treatment, as well as low levels of expression of collagen types I and X were observed in the defect area, whereas in the control group without treatment, the results were opposite.	[125]
(not specified)	Cynomolgus monkey cells (not specified)	Chondrocyte-specific iPSC reporter lines were created by introducing COL11A2-EGFP human transgene. Differentiation through the stage of MSC-like precursors using Wnt3a and Activin A. Then, differentiation was carried out using ascorbic acid, BMP2, TGFβ1, and GDF5; after 14 days of cultivation, there was a transfer to a suspension culture.	-	The organoids stained positively for safranin O, and immunostaining detected the presence of large amounts of type II collagen. Type I collagen was found only on the periphery of the organoid.	Cynomolgus monkey, transplantation of cell organoids into chondral defects in the femoral trochlear crest of the right knee joints	Four weeks after transplantation, the defect was filled with transparent hyaline-like tissue, and at week 17, white cartilaginous tissue was observed. Allogeneic organoid transplantation did not elicit an immune response in primates. Positive safranin O staining was observed at both 4- and 17-weeks post-transplant, indicating cartilage matrix production. scRNA-seq showed that almost all cells in transplanted organoids expressed COL2A1 but not COL1A1. Cells in post-transplant organoids were identical to native chondrocytes by cluster analysis, excluding cells associated with integrin signaling.	[59]

The conditions and duration of cultivation are also noteworthy factors affecting the quality of iPSCs in general and the number of their epigenetic markers in particular. A decrease in the number of differentially methylated regions of iPSCs with different origins increases the ability of the line to differentiate with the same efficiency in any of the three germ sheets [76]. There is also evidence that epigenetic features left over from the original cells are lost during prolonged cell passages. However, the extent to which such a phenomenon is observed in the case of using iPSC cells obtained from the tissue cells of pathology patients for differentiation has not been practically studied [92].

iPSC differentiation protocols are also extremely important for obtaining cell cultures with characteristics as close as possible to native chondrocytes. It was demonstrated that iPSCs from OA patients finally differentiated by the chondrogenic pathway were no longer significantly different from the control group [67]. This suggests that the quality of differentiation can directly affect the phenotype and gene expression in such cells. Some researchers also believe that in order to elucidate the chondrogenic ability of iPSCs from OA patients it is necessary to optimize differentiation protocols, since insufficient production of cartilage markers may be associated not with pathology but with shortcomings in in vitro chondrogenesis [64]. To date, many protocols for chondrogenic differentiation have been developed. Some of them show high efficiency in both in vitro [119,121,126] and in vivo [57,58,59] experiments. In some protocols, researchers resort to techniques that reduce some risks, such as suppression of hypertrophy, as it is possible to suppress the expression of COL10A1 [91]. To reduce the risk of off-target differentiation, one of the latest protocols proposed using small molecule inhibitors of WNT and MITF [118]. In most studies, TGFβ1 and TGFβ3, as well as BMP2 and BMP4, are used as the main chondrogenic inducers. They showed that lentiviral transduction of TGFβ1 that ensures the constitutive synthesis of this factor in cells, in combination with co-cultivation with mature cartilage tissue, enhanced in vitro chondrogenesis and contributed to a decrease in VEGF expression, including through factors secreted by mature chondrocytes, for example metastatin [64]. To differentiate the bone marrow MSCs of healthy donors in the chondrogenic direction, constitutive production of BMP2 was also achieved by means of vector transduction with this protein, which proved effective for further chondrogenesis [127]. Perhaps this approach can also be useful for improving the chondrogenic differentiation of iPSCs in patients with joint pathologies. It was also reported that hypoxia had a positive effect on chondrogenic differentiation and contributed to an increased proportion of SOX9-positive cells in the culture [128].

A necessary step to evaluate the functionality of the resulting tissue-engineered structure based on chondrocytes is a biomechanical function study, including compression and shear tests, a measurement of surface roughness, and a friction test. It has been shown that the structures obtained from the iPSCs of healthy patients are superior in characteristics to those from the chondrocytes obtained from pathology patients, which makes them promising for cartilage tissue engineering and the therapeutic restoration of focal defects [129]. In addition, various matrices are widely used to improve the mechanical properties of cartilage tissue in vitro, as well as to increase the efficiency of chondrogenesis by creating a microenvironment. Thus, alginate [65], polycaprolactone [89], and collagen [129] matrices were used to differentiate iPSCs from OA patients. The effectiveness of bioceramic scaffolds with lithium ions was also reported, and their use accelerated the differentiation process and prevented the hypertrophy of newly differentiated chondrocytes from the iPSCs of healthy donors [89]. In addition, with the help of various scaffolds in the approximation, it is possible to recreate spatiotemporal signals simulating native chondrogenesis [130]. External mechanical influences can also contribute to better chondrogenesis, since mechanotransduction mechanisms that trigger biochemical signals in chondrocyte precursors in response to physical forces play an important role in the process of chondrogenic differentiation in native cartilage [131]. Thus, Limraksasin et al., demonstrated the positive effect of dynamic conditions of 0.3 and 0.5 Hz shaking on chondrogenesis in vitro. At the same time, shaking the culture increased the expression of the TGFβ1, TGFβ2, and TGFβ3 and Wnt3a, Wnt5a, and Wnt5b genes involved in the modulation of proliferation and chondrogenesis [132]. In another study, they also showed the effect of mechanical stimulation on chondrogenesis through the BMP4–Smad signaling pathway [133]. Later, this group of authors proposed an efficient protocol combining the use of a piggyBac-vector-based tet-controlled BMP-4 gene regulation system for iPSCs and 3D shaking culture. Differentiated 3D pellets showed better production of cartilage matrix compared with static culture and restored osteochondral defects after transplantation to rats in an in vivo model without tumor formation and teratomas [124].

Despite the advances in research on iPSC chondrogenesis over the years and the bright prospect of this cellular resource for the treatment of articular cartilage diseases, there is still no clinical application. Including all the aforementioned optimization approaches, researchers still have a lot of problems to solve. Thus, despite the existence of many protocols for in vitro chondrogenesis, there are still no generally accepted protocols approved for use in clinical trials [134]. Although the protocol proposed by Yamashita et al., has demonstrated satisfactory performance in vitro and in vivo over the years [59,121], having options in the choice of protocols for differentiation would improve the situation. Despite the fact that reports of recent chondrogenic differentiation protocols do not mention the formation of tumors and teratomas, the safety aspect still requires particularly tight control. The ganciclovir tyrosine kinase–ganciclovir (HSV-TK/GCV) and induced caspase 9 (iCasp9) systems, as well as R-17F antibodies, are effective for the selective elimination of undifferentiated or misdifferentiated cells [114,116,117]. In this review, we draw attention to the importance of choosing a cellular source for reprogramming, as there is evidence of a difference in the chondrogenic potential of iPSCs from healthy and OA donors, which may become an obstacle to their widespread use and the creation of common cell banks [64,65,66,67,85,87]. In addition, data are emerging on genetic and epigenetic differences between native [135] and iPSC-derived [85] chondrocytes from patients with OA and healthy patients, mainly associated with regulatory pathways, metabolic processes, and epigenetic gene regulation. However, the mechanisms linking the found differences with the efficiency of chondrogenesis are still unknown. It is also necessary to check whether the qualitative features of iPSCs and their chondrogenic derivatives from patients with OA are preserved over a large number of passages. Therefore, in future studies it will be necessary to conduct an extensive comparative analysis of the sequences, epigenetic marks, and expression of healthy and OA cells to identify potential regulatory levers; this will help to better understand the development of pathology and the ability to improve in vitro chondrogenesis.

## 6. Conclusions

Although the subject of using autologous iPSCs in patients with various joint diseases and their derivatives in clinical practice has been little studied, we can use the available work to summarize the general state of affairs in this area today. We have conflicting data on the phenotypes of chondrocytes differentiated from iPSCs from different research groups. In some studies, normal chondrogenesis was reported, accompanied by the expression of chondrogenic markers and the production of a cartilage matrix. However, there is also evidence of abnormal variants of the phenotype or incorrect production of cartilage matrix and degeneration marker expression. Nevertheless, all iPSC lines in the reviewed articles met the standard criteria of pluripotency, as they expressed the necessary markers and differentiated into three germ sheets. Perhaps measures optimizing cultivation and differentiation in order to enhance chondrogenesis will help solve the problem of imperfect chondrogenesis. However, in order to obtain a safe cell product for clinical use, a significant amount of work will still be required using modern genetic and omics methods for the comparative evaluation and characterization of pluripotent stem cells and their differentiated derivatives, which will allow for the maximum efficiency of developments in the field of cellular technologies.

## Figures and Tables

**Figure 1 ijms-24-14408-f001:**
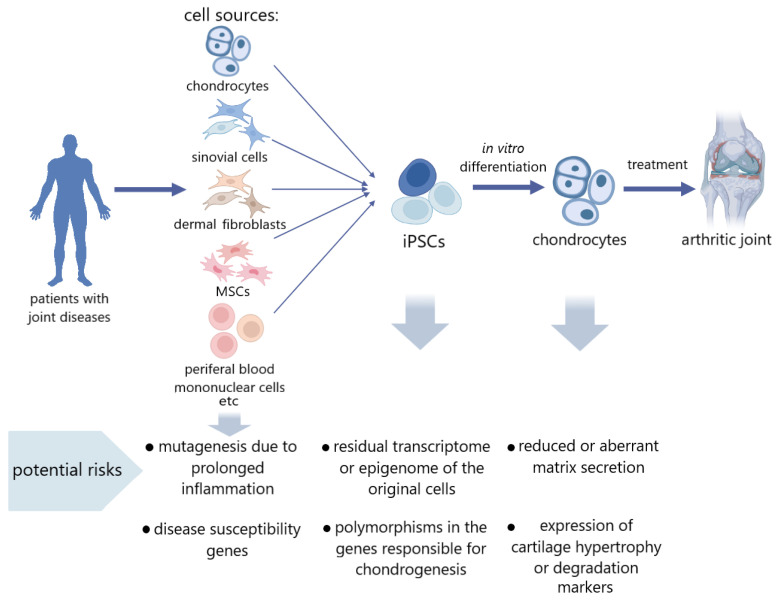
Potential risk factors in the use of donor cells with joint diseases for tissue engineering.

**Table 1 ijms-24-14408-t001:** Current progress in obtaining iPSCs and their chondrogenic derivatives from various initial somatic cells of joint disease patients.

Reprogramming Method	Type of Reprogrammed Cells, Pathology	Comparison with Healthy Donor Cells	iPSC Characteristics	Method of Chondrogenic Differentiation	Characteristics of Chondrogenic Derivatives	Link
Sendai virus	Fibroblasts of the skin, non-erosive OA of the hand, risarthrosis of the arm	+	Alkaline phosphatase activity; expression of OCT4, SOX2, KLF4 and C-MYC, as well as NANOG and CRYPTO. Ability to differentiate into cells of three germ leaves in embryoid bodies (EBs): expression of α-fetoprotein (endoderm), α-actin of smooth muscles (mesoderm), neuron-specific β-tubulin class III (ectoderm). After reprogramming, allelic variants of SNP in the GDF 5, SMAD3, ALDH1A2, and IL 1R1 genes observed in parental fibroblasts were preserved. There were no significant differences from the IPSC of healthy donors.	Directed differentiation in EBs using Wnt3a, Activin A, then BMP2, GDF5, TGFβ3.	Staining with Masson’s trichrome and safranin O showed a lower presence of the collagen matrix and proteoglycans, respectively, compared with chondro-derivatives of healthy donors.	[64]
Sendai virus	MSCs, OA	+	Alkaline phosphatase activity; expression of OCT4, SOX2, NANOG, SEA4. Analysis of teratoma formation demonstrated differentiation into three germ leaves: pulmonary epithelium (endoderm), embryonic mesenchyme, smooth muscle cells, adipose, cartilage and bone tissue (mesoderm), neural tube, horny epidermis (ectoderm). There were no significant differences from the iPSCs of healthy donors.	-	-	[81]
Lentivirus system	Chondrocytes, OA	+	Alkaline phosphatase activity, expression of OCT4, SOX2, NANOG, KLF4, TRA-1–60; however, klf4 expression was lower in iPSCs from both healthy and OA chondrocytes compared with other pluripotency genes.	Differentiation through the stage of MSC-like precursors using FBS and bFGF. Then differentiation was carried out in 3D pellet conditions using BMP2.	On days 4, 7, and 21 of differentiation, SOX9, COL2A1, ACAN, and PRG4 mRNA expression was significantly higher in iPSC derivatives from healthy donors compared with iPSC derivatives from OA donors. The micromass culture of iPSC derivatives from healthy donors was intensely stained with alcian blue, in contrast to iPSC derivatives from donors with OA. At the stage of MSC-like progenitors, iPSC derivatives from donors with OA showed significantly higher levels of expression of the pro-inflammatory genes CCL2, CCL3, CXCL3, and NOS2 in the aggressive environment of the IL1β or TNF-α inflammatory stimulus compared with iPSC derivatives from healthy donors.	[85]
Sendai virus	PBMCs, fibroblast-like synoviocytes, RA	−	Typical iPSC morphology, expression of OCT4, NANOG, TRA-1–81, and SSEA-4. Ability to differentiate into cells of three germ leaves in ET: expression of NF, NESTING, TUBB3 (ectoderm), vimentin, BRACHYURY T, NKX 2.5 (mesoderm), GATA4, SOX17, FOXA2 (endoderm).	-	-	[90]
Episomal plasmid vectors without transgenes	PBMCs, ankylosing spondylitis	−	Typical morphology of ESC, expression of OCT4, SEA 4, SOX2, and TRA-1-60. Analysis of teratoma formation showed the ability to differentiate into three germ leaves: nervous (ectoderm), cartilaginous (mesoderm), glandular (endoderm) tissue.	-	-	[89]
Lentivirus system	Fibroblast-like synoviocytes, OA, RA	−	Expression of NANOG, OCT4, SOX2, KLF4, TRA-1-80, TRA-1-60, REX, and SSEA-4; however, before reprogramming, the expression level of KLF4 was high. Analysis of teratoma formation showed the ability to differentiate into three germ leaves: skin structure (ectoderm), blood vessels and adipose tissue (mesoderm), gland (endoderm).	-	-	[87]
Lentivirus system	Fibroblast-like synoviocytes, RA	−	Alkaline phosphatase activity, expression of OCT3/4, SOX2, NANOG, LIN28, DPPB5, and TDGF1, as well as SEA 4, TRA-1-60, TRA-1-81, and Klf4. Analysis of teratoma formation showed the ability to differentiate into three germ leaves: the formation of glandular and adipose tissues and blood vessels. Expression of OTX2 (ectoderm), BRACHYURY (mesoderm), SOX17 (endoderm) was observed.	-	-	[66]
Sendai virus	Dermal fibroblast, OA fingers with early onset	+	Alkaline phosphatase activity, expression of SEA 4, TRA-1-60, TRA-1-81, LIN28, OCT4, SOX2, and KLF4. Expression of OCT4 and NANOG was higher in the iPSCs obtained from an OA patient compared with the iPSCs of a healthy donor.	Directed differentiation using TGFβ1, production of chondrogenic pellets and cultivation in a medium with the addition of dexamethasone, TGFβ3.	Pellets from the iPSC of an OA patient had a larger size and vacuum-like formations inside the structures. On both days 7 and 21, expression of the SOX9 chondrogenic marker in the iPSC derivatives of the OA patient was high, though a slight tendency to decrease expression was recorded. Expression of ASAT was low, both in the derivatives of the patient and the healthy donor on days 7 and 21. In addition, on days 7 and 21 COL2A1 expression was significantly lower than in IPSC derivatives of a healthy donor. On day 7, a higher expression of COL1A1 was recorded in the cells of an OA patient than in the cells of a healthy donor, but no difference was recorded on day 21. Differences in the expression of the COL10A1 hypertrophy marker were also not observed. On day 7, VEGF expression was higher than in the control group, but on day 21, low expression rates were observed. AQP1 was expressed significantly more than in IPSC derivatives of a healthy donor.	[67]
Retrovirus system	MSCs-like synovial cells, OA	+(hESCs)	iPSC-like morphology, alkaline phosphatase activity, expression of OCT-4, NANOG, SOX2, hTERT, RES 1, LIN28, TDGF, TRA-1-60, and SSEA-3. Analysis of teratoma formation in vivo and EBs in vitro demonstrated a decrease in the expression of OCT-4 and NANOG pluripotent markers, as well as the ability to differentiate into three germ sheets: expression of Pax6, Tuj1, and Nestin (ectoderm), Brachyury, GATA-2, desmin, and α-actin of smooth muscles, a number of chondrogenic markers, such as SOX9, ACAN, COL2 (mesoderm), GATA-6, SOX17, FoxA2, and α-fetoprotein. There were no significant differences from the indicators of hESC expression. The reprogramming efficiency was 0.007–0.01%.	Directed chondrogenic differentiation in EBs (protocol not specified), the production of chondrogenic granules (protocol not specified), the use of an agarose substrate, as well as a three-dimensional polycaprolactone scaffold.	Cartilage-like cell aggregates were formed that stained positively with alcian blue and safranin O. Expression of SOX9, aggrecan, and type II collagen was also observed. Expression of type X collagen in the differentiated derivatives of one of the two iPSC lines obtained was higher than in the hESC derivatives. Aggrecan expression was higher in the derivatives of both iPSC lines compared with the derivatives of hESCs. Cells cultured on a three-dimensional scaffold for 2 months showed the morphology of chondrocytes, intense expression of SOX9 and collagens of types I, II, and X, and were stained with alcian blue.	[88]
Lentivirus system	Chondrocytes, OA of knee joins	+	Typical morphology, alkaline phosphatase activity, expression of OCT-4, SOX-2, REX-1, NANOG, SSEA-1, SSEA-4, TRA1-60, and TRA1-81 were observed in two of the three obtained colonies. ET analysis demonstrated the ability to differentiate into derivatives of three germ leaves: Nestin expression, histological characteristics of nervous tissue (ectoderm), desmin, histological characteristics of bone and muscle tissue (mesoderm) and α-fetoprotein, histological characteristics of intestinal-like and respiratory-like epithelium (endoderm). The clones were directionally differentiated into derivatives of three germ leaves in vitro: expression of NSE, NF-M, MBP, GAD, Nestin (ectoderm), GATA-4, NKX2.5, MLC-2A and MLC-2V (mesoderm), PDX-1, PAX-6, NKX2.2, and insulin (endoderm).	Transfection with a TGFβ1-carrying lentivirus for endogenous expression, use of an alginate matrix, and co-cultivation with native mature chondrocytes.	Increased expression of TGFβ1 in the transduced iPSCs was confirmed by Western blotting. In the experimental group of TGFβ1-induced iPSCs in the co-culture alginate matrix, the expression of type II collagen, aggrecan, and COMP was significantly higher than in other experimental groups but lower than in native chondrocytes. VEGF expression was zero.	[65]
mRNA transfection	Chondrocytes, skin fibroblasts, foreskin fibroblasts, OA (after replacement therapy with autologous chondrocytes)	+(ESCs)	Alkaline phosphatase activity, expression of OCT4, SSEA 4, TRA-1-60, NANOG. Reprogramming efficiency is approximately 0.1%. ET analysis demonstrated the ability to differentiate into derivatives of three germ leaves: βΙΙΙ-tubulin (ectoderm), α-actin of smooth muscles (mesoderm), HNF3ß (endoderm). Teratoma analysis also demonstrated the formation of a neural epithelium (ectoderm), spontaneously contracting cardiomyocytes (mesoderm), cylindrical epithelium (endoderm). No significant differences from the ESC were recorded.	Directed differentiation in monolayer culture in the DEF-CS system using Activin-A, Wnt3a, FGF2, BMP4 at the first stage, FGF2, BMP4, follistatin, and NT4 at the second, FGF2, BMP4, NT4, and GDF5 [91]. Preparation of chondrogenic granules, directed differentiation using TGFβ1 and dexamethasone.	Expression of pluripotency markers decreased with differentiation, expression of CDH1, MLH1, and GSC showed the mesodermal direction of differentiation. High expression of PDGFR, SOX6, SOX9, ACAN, COL2A1 types A and B. Expression of type X collagen in the late differentiated derivatives was low.	[73]

## Data Availability

Not applicable.

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
