# Peer review of "Clinical Potential of Cellular Material Sources in the Generation of iPSC-Based Products for the Regeneration of Articular Cartilage"

_ijms, 2023, doi:10.3390/ijms241914408_

Round 1

Reviewer 1 Report

Please see comments attached

Minor editing of English language required

Author Response

Dear reviewer,

Thank you for your valuable comments and observations. We tried to take them into account when editing the manuscript.

(Due to the addition of text to the manuscript, the line numbering has changed)

  1. Please provide references for lines 39-42, 42-44, 56-59, every sentence in paragraphs in lines 63-79 (currently, it all just sounds like statements with no evidence or backing for what you are saying). Also for lines between 81-98, line 232, 235-237, 238-239, 268-269

References for lines 39-42, 44-46, 57-58, 65-81, 82-110, 92-93, 139-142, 272, 275-277, 278-279, 313-316 were added

  1. Please re-write line 54 as ‘Non-surgical approaches include pharmacological and non-pharmacological approaches. Non-pharmacological approaches include…’ and the continue with what you have written

 The line 56 was rewritten according to the recommendation

  1. Please check the spelling of ‘gonarthrosis’ throughout your text

Spelling was corrected in line 62

  1. There are many other sources of chondrocytes, the major one being bone marrow mesenchymal stem cells (MSCs). There is plenty data regarding this subject. While you need not write about them, you must acknowledge these cells as a major source with recent references in your manuscript

MSCs and autologous chondrocytes data were added (lines 92-102)

  1. For lines 113-115, your mention ‘several studies’ but mention just one reference (number 16). Please add 3-4 relevant studies to match your statement

Three more references were added (line 129-130)

  1. For line 123-126, you mention ‘there are also publications on….’ And have no references to support any of these publications at the end of the sentence. Please correct this throughout your manuscript – for an article to be acceptable, you need to provide more references. Usually 1 or 2 references every 2-3 sentences to support what you are saying.

Four references were added (lines 139-142)

  1. Please check the font of the text in the figure

The font was fixed

  1. Please consider re-arranging your manuscript with an ‘Epigenetic section’ given that you discuss this is detail

We've changed the title of the first section to focus on epigenetics and added more information (lines 230-235, 248-264)

  1. It will add to our manuscript if you added a table of current clinical trials on iPSCs for cartilage regeneration in arthritis. This information would strengthen your paper

Table 2 «Recent advances in chondrogenic differentiation of iPSCs and models in vivo» was added

  1. Please format your table in the landscape format and increase the width of columns that have more information to better fit in the details and to make it neat

Table 1 was formatted

  1. Please include ‘challenges, conclusions and future directions’ as the last section. Clearly outline the challenges faced by iPSCs for cartilage remodelling based on your manuscript and remind the readers that it is not an easy task. What other challenges do you foresee in this field? Donor derived cells may indicate donor variations? Any others? For future directions – suggest which of the genes that you have discussed should be included and why in future investigations.

Discussion was added (lines 522-548)

  1. Regarding your title – you focus only on arthritis and not on ‘medicine’ on the whole. Please change your title to match your manuscript

We suggested a different title “Clinical potential of cellular material sources in the generation iPSC-based products for the regeneration of articular cartilage”

Reviewer 2 Report

The authors of the article have described the effect of iPSC cells for chondrogenic differentiation in the treatment of osteoarthritis (OA). Specifically, they focus on the generation of iPSC cells from osteoarthritic patients and show the inconclusive results from these patients compared from cells from non-OA patients.

This is an important and interesting review for the readers. The following points need to be considered in their revision,

1.       Osteoarthritis is a multi-factorial that is not the sole preserve of persons within the high risk group, described in this case, as athletes, military personnel etc.. Aging and diet are risk factors for the disease and this needs to be stated in the introduction sections of the manuscript.

2.       There is no discussion about the regulations that do not permit the use of iPSC generated chondrocytes for cell therapies. The authors should discuss this and explain how these regulations can be overcome in the future.

3.       There is a lack of references in the introduction for the statement made and this needs to amended-

4.       A discussion or a table describing the advantages and disadvantages of iPSCs compared to other adult stem cells (e.g. mesenchymal stem cells).

5.       In the opinion of the authors, what is the appropriate cells to be utilised from OA patients for creating iPSCs for cartilage repair or is the underlying epigenetics controlling the chondrocyte process irreversible as stated in the current review ? More opinion on the current literature should be included in the review.

6.       Are there publication with stable iPSC generated chondrocytes and have these been used in clinical application ? There remains conjecture on creating stable chondrocytes from iPSCs and thus section on how this induced in healthy tissues is needed and information on whether there are clinical trials using iPSCs for OA would of interest to readers.

7.       Apart from hypoxia, what other environmental stimuli can be used to stablise or potentially recover the iPSC chondrocyte phenotype generated diseased cartilage tissues ? Is there evidence that mechanobiology can help recover the phenotype or similar stimuli.

8.       Senescent cells are known to be present with OA tissues and potentiate the degenerative process. How does the senescent phenotype affect the induction of iPSCs ? Similar to question 5, could this have affected the generation of stable chondrocytes from iPSCs.

Author Response

Dear reviewer,

Thank you for your valuable comments and observations. We tried to take them into account when editing the manuscript.

(Due to the addition of text to the manuscript, the line numbering has changed)

  1. Osteoarthritis is a multi-factorial that is not the sole preserve of persons within the high risk group, described in this case, as athletes, military personnel etc.. Aging and diet are risk factors for the disease and this needs to be stated in the introduction sections of the manuscript.

Statement was added (lines 42-44)

  1. There is no discussion about the regulations that do not permit the use of iPSC generated chondrocytes for cell therapies. The authors should discuss this and explain how these regulations can be overcome in the future.

Discussion was added (lines 522-549)

  1. There is a lack of references in the introduction for the statement made and this needs to amended-

Refferences were added (lines 39-42, 44-46, 57-58, 65-81, 82-110)

  1. A discussion or a table describing the advantages and disadvantages of iPSCs compared to other adult stem cells (e.g. mesenchymal stem cells).

Discussion was added (lines 92-102, 111-128)

  1. In the opinion of the authors, what is the appropriate cells to be utilised from OA patients for creating iPSCs for cartilage repair or is the underlying epigenetics controlling the chondrocyte process irreversible as stated in the current review? More opinion on the current literature should be included in the review.

In our review, we focused on the impact of cartilage pathologies on a potential cellular product of regenerative medicine, it was not our goal to evaluate different cellular sources for iPSCs, in addition, each source has a place depending on the goals and capabilities of the researchers. With regard to epigenetics, at the moment it is impossible to unambiguously speak of its reversibility or irreversibility. With regard to epigenetics, at the moment it is impossible to speak unambiguously about its reversibility or irreversibility due to the small number of studies, but we refer to work that considers the possibility of influencing epigenetics in the future (lines 248-262, 538-548).

  1. Are there publication with stable iPSC generated chondrocytes and have these been used in clinical application? There remains conjecture on creating stable chondrocytes from iPSCs and thus section on how this induced in healthy tissues is needed and information on whether there are clinical trials using iPSCs for OA would of interest to readers.

Table 2 «Recent advances in chondrogenic differentiation of iPSCs and models in vivo» was added

  1. Apart from hypoxia, what other environmental stimuli can be used to stablise or potentially recover the iPSC chondrocyte phenotype generated diseased cartilage tissues? Is there evidence that mechanobiology can help recover the phenotype or similar stimuli.

Biomechanics impact data were added (lines 507-521)

  1. Senescent cells are known to be present with OA tissues and potentiate the degenerative process. How does the senescent phenotype affect the induction of iPSCs? Similar to question 5, could this have affected the generation of stable chondrocytes from iPSCs.

Discussion was added (lines 399-423)

Round 2

Reviewer 1 Report

Please see document attached

Minor editing of English language required

Author Response

Dear Reviewer,

We thank you again for all your comments and recommendations. We tried to correct spelling and grammatical errors in the text, and added reference-links to the articles as you recommended (line 401).

Reviewer 2 Report

The authors have addressed my points appropriately.

Author Response

Dear Reviewer,

We are grateful to you that the changes made to the manuscript according to your comments made our publication better.